# Enhancing Thermal Conductivity in Polymer Composites through Molding-Assisted Orientation of Boron Nitride

**DOI:** 10.3390/polym16081169

**Published:** 2024-04-21

**Authors:** Yongjia Liu, Weiheng Gong, Xingjian Liu, Yicheng Fan, Aihua He, Huarong Nie

**Affiliations:** Shandong Provincial Key Laboratory of Olefin Catalysis and Polymerization, Key Laboratory of Rubber-Plastics (Ministry of Education), School of Polymer Science and Engineering, Qingdao University of Science and Technology, Qingdao 266042, China; liuyongjia1998@foxmail.com (Y.L.);

**Keywords:** thermal conductivity, filler orientation, compression molding, polymer composites

## Abstract

Incrementing thermal conductivity in polymer composites through the incorporation of inorganic thermally conductive fillers is typically constrained by the requirement of high filler content. This necessity often complicates processing and adversely affects mechanical properties. This study presents the fabrication of a polystyrene (PS)/boron nitride (BN) composite exhibiting elevated thermal conductivity with a modest 10 wt% BN content, achieved through optimized compression molding. Adjustments to molding parameters, including molding-cycle numbers, temperature, and pressure, were explored. The molding process, conducted above the glass transition temperature of PS, facilitated orientational alignment of BN within the PS matrix predominantly in the in-plane direction. This orientation, achieved at low filler loading, resulted in a threefold enhancement of thermal conductivity following a single molding time. Furthermore, the in-plane alignment of BN within the PS matrix was found to intensify with increased molding time and pressure, markedly boosting the in-plane thermal conductivity of the PS/BN molded composites. Within the range of molding parameters examined, the highest thermal conductivity (1.6 W/m·K) was observed in PS/BN composites subjected to five molding cycles at 140 °C and 10 MPa, without compromising mechanical properties. This study suggests that compression molding, which allows low filler content and straightforward operation, offers a viable approach for the mass production of polymer composites with superior thermal conductivity.

## 1. Introduction

As the density of electronic components and power density continue to escalate, thermal management has emerged as a crucial determinant of performance stability and the longevity of electronic devices [1,2]. Polymer-based thermally conductive materials, known for their excellent electrically insulating attributes, light weight, and ease of processing, have begun to supplant traditional metal-based counterparts, demonstrating significant potential in thermal management applications [3,4]. However, polymers are inherently thermally insulating, and enhancing their thermal conductivity involves either augmenting the intrinsic thermal conductivity of the polymers or incorporating high-thermal-conductivity fillers [5,6,7,8,9,10].

Enhancement of the intrinsic thermal conductivity of polymers necessitates the reorganization of molecular chains into highly crystalline structures by modifying the polymer chain architecture [11,12,13,14,15,16], an example being the utilization of AFM probes [15]. This approach enables polymers to achieve substantially high thermal conductivities in the direction of orientation. However, this method is prohibitively expensive and complex, presenting challenges for large-scale applications. 

In contrast, integrating thermally conductive fillers into a polymer offers a more feasible strategy for attaining high thermal conductivity in these materials. Notably, the inclusion of thermally conductive but electrically insulating fillers preserves the outstanding electrically insulating properties of polymers [3,4]. However, electrically insulating fillers primarily rely on phonon heat conduction and necessitate higher filler concentrations to establish a continuous thermally conductive network and minimize interface thermal resistance, thereby enhancing thermal conductivity. The employment of linear or planar fillers [17] with high aspect ratios, such as boron nitride nanosheets [4,18,19,20,21,22,23,24,25,26] (BNNSs), silicon carbide (SiC) nanowires [27], graphene or graphene oxide (GO) [28,29,30,31,32,33,34,35], carbon nanotubes (CNTs) [36,37,38,39], and MXene [8,40], facilitates the creation of a thermally conductive network at relatively low filler loadings compared to other filler geometries. Furthermore, the typical association of high filler loadings with compromised mechanical properties has led researchers to modify filler surfaces to enhance their dispersion within polymer matrices, thereby realizing highly thermally conductive polymer composites at reduced filler concentrations [26,36,41,42,43,44,45,46]. However, these surface modification processes often entail intricate chemical reactions that may diminish the fillers’ intrinsic thermal conductivity. 

Recently, the development of filler networks has gained prominence in various environmental fields, underscoring a paradigm shift towards innovative composite materials [26,36,41,42,43,44,45,46]. Guo et al. [47] utilized a gradient temperature field to fabricate alternating layers of GO and BN through an ice-template method, achieving a polyimide composite with an in-plane thermal conductivity of 1.49 W/m·K at a filler loading of 3.8 wt%. Wu et al. [48] engineered a thermal pathway combining micro-scale BN and nano-scale alumina within poly(dimethylsiloxane) using an electric field, resulting in an in-plane thermal conductivity of 0.228 W/m·K. Yuan et al. [23] aligned FeCo-BN vertically within poly(dimethylsiloxane) under magnetic field, obtaining a thermal conductivity of 2.25 W/m·K with a filler composition of 30 wt% FeCo and 50 wt% BN. Beyond electric and magnetic fields, mechanical force fields have garnered significant interest as a way of establishing thermally conductive pathways due to their operational simplicity and accessibility. Chen et al. [31] prepared a pea-pod-like structure of aluminum oxide–graphene foam under directed gas pressure, employing 42.4 wt% spherical alumina as support and 12.1 wt% sheet-like graphenes as a filler, to produce epoxy resin composites exhibiting an in-plane thermal conductivity of 33.4 W/m·K. Through the application of shear fields within an extruder, Zhang et al. [49] oriented BN within high-viscosity, high-density polyethylene, achieving an anisotropic index of 480% and an in-plane thermal conductivity of 3.57 W/m·K at a 40 wt% loading rate. Yu et al. [50] achieved an in-plane thermal conductivity of 50.3 W/m·K in thermoplastic polyurethane composites by ball milling and hot pressing of hexagonal boron nitride into the polymer matrix. However, the pre-construction of filler pathways may encounter challenges from the injection of polymer melts or fluids, potentially compromising the integrity of these pathways. Additionally, the equipment required for such methodologies is not readily scalable and accessible for mass production, and the employment of electric and magnetic fields often necessitates additional reagents for filler modification, presenting further obstacles to widespread application.

In this study, we developed a polystyrene (PS)-based composite with enhanced in-plane thermal conductivity by facilitating the orientation of BN through molding. The impact of various molding parameters, including the molding temperature, cycle numbers, and pressure, on the BN’s orientation within the PS matrix and the resultant thermal conductivity of the PS/BN composites was meticulously evaluated. Through parameter optimization, it was demonstrated that these PS/BN composites, when subjected to five molding cycles at 140 °C and 10 MPa, can achieve an in-plane thermal conductivity of 1.58 W/m·K with only 10 wt% BN loading. This study introduces a straightforward method for producing highly thermally conductive polymer composites, which are promising for large-scale and continuous production due to their accessible processing techniques and minimal filler requirements. 

## 2. Materials and Methods

### 2.1. Materials

Hexagonal boron nitride (h-BN, purity > 98.5%, 1 μm) was purchased from Shanghai Maclin Biochemical Technology Co., Ltd. (Shanghai, China). Polystyrene (PS, general type I) was procured from Aladdin Reagents Co., Ltd., Shanghai, China. Xylene and anhydrous ethanol were obtained from Sinopharm Chemical Reagents Co., Ltd. (Shanghai, China). All reagents were used as received.

### 2.2. Fabrication of Thermal Conductive PS/BN Composites

The fabrication process commenced with the production of granular PS/BN materials through a solution mixing technique. Initially, BNNSs were generated by dispersing a predefined quantity of BN powder in xylene via ultrasound for 1 h at room temperature. Subsequently, PS particles were incrementally introduced to the BNNS suspension under continuous stirring at 90 °C until completely dissolved. The addition of anhydrous ethanol precipitated the solids, which were then subjected to multiple washes with anhydrous ethanol and dried under a vacuum at 40 °C for 24 h, yielding granular PS/BN materials with BN loadings of 10 wt% and 30 wt%, respectively. Following this, PS/BN composites exhibiting high thermal conductivities were produced by compression molding. Specifically, 6 g of the PS/BN granular material, containing 10 wt% BN, was positioned in a mold of dimensions 85 × 115 × 0.5 mm. This was transformed into a plastic sheet through 10 min of compression molding under predetermined parameters. The resultant plastic sheet underwent additional molding cycles (one, three, and five cycles) to generate a series of molded PS/BN composites by varying molding parameters, as delineated in Table 1. For comparative purposes, control PS/BN samples were fabricated by casting and drying the PS/BN/xylene solution in a mold.

### 2.3. Characterization

Field-emission scanning electron microscopy (SEM, JSM-7500-F, JEOL, Tokyo, Japan) operating at an acceleration voltage of 3–5 kV was utilized to examine the dispersion and alignment of BNNSs within the PS matrix. The orientation of the BN within the PS was quantitatively assessed via X-ray diffraction (XRD) patterns obtained from the PS/BN composites by employing a XRD instrument (Ultima IV, Rigaku, Tokyo, Japan) at a scan rate of 10°/min. The thermal diffusivities of the PS/BN composites were determined using the laser flash analysis method (LFA 467, NETZSCH, Selb, Germany). Samples were meticulously prepared into 25 mm diameter circular chips, with both surfaces uniformly coated with a thin graphite powder layer to facilitate absorption and dispersion of the laser pulse. The analysis incorporated in-plane and pulse correction techniques with a laser voltage of 180 V and a pulse width of 0.1 ms. Thermal conductivity (*K*) was calculated using the following formula: *K = α × ρ × C_p_*
where *α*, *ρ*, and *C_p_* are the thermal diffusivity (mm^2^/s), density (g/cm^3^), and specific heat (J/kg·K), respectively. The heat capacity and density of the samples were ascertained using a DSC8500 instrument (PerkinElmer, Waltham, MA, USA) at a heating rate of 10 °C/min and a density balance (METTLER TOLEDO, Zurich, Switzerland), respectively. The reported thermal conductivity values represent the average of three samples. Additionally, the samples were placed on a hot bench at 60 °C and then removed from the bench. The heating and cooling processes of the samples were recorded using thermal imaging (Uti 260D) to evaluate the warming and heat-dissipation characteristics of the sample chips. Thermogravimetric analysis (TGA) involved heating materials from ambient temperature to 600 °C in an oxygen atmosphere at a rate of 10 °C/min to assess thermal stability. The mechanical properties of the PS/BN composites were determined using a Zwick BTE-TC02.00 (Ulm, Germany) material testing apparatus.

## 3. Results and Discussion

PS/BN composites containing 10 wt% BN were fabricated by compression molding following the parameters detailed in Table 1. The molding temperatures were maintained above the glass transition temperature of PS to ensure adequate mobility of the PS chains, thereby facilitating BN orientation. Prior to subsequent compression molding cycles, the PS/BN composites were cut and folded from the periphery to the center to maintain uniform BN loading and thickness across samples. It was posited that the compression molding process would orient the planar fillers perpendicular to the direction of pressure, thus establishing a thermally conductive pathway at minimal filler loading and significantly enhancing the in-plane thermal conductivity of the polymer composites (Figure 1).

For all molded PS/BN composites fabricated, thermal stability was rigorously assessed to confirm that the selected molding conditions did not adversely affect the integrity of PS. The TGA results, illustrated in Figure 2, demonstrated two stages of weight loss for all the PS/BN composites. Stage I involved the removal of impurities and solvent residues, and stage II started from 270 °C, corresponding to the decomposition of PS. This outcome indicated that the molding temperatures employed were within a safe processing window, regardless of the number of molding cycles applied. In contrast, the initial decomposition temperature (270 °C) of the control PS/BN sample was observed to be marginally higher than that (250 °C) of pure PS, suggesting that the inclusion of BN contributes to the enhanced thermal stability of the composite. Notably, after molding was performed at temperatures of 120 °C, 140 °C, and 160 °C for five cycles, the decomposition temperatures of the molded PS/BN composites remained comparable to those of the control samples. This trend persisted even for composites subjected to more extensive molding conditions at higher temperatures, such as 180 °C, 200 °C, 220 °C, and 240 °C, underscoring the reliability and practicality of the molding process in generating polymer composites with superior thermal conductivities.

Given the widespread acknowledgment that high filler loadings can compromise mechanical integrity, the tensile strength of the molded PS/BN composites prepared herein was evaluated against both the control PS/BN samples and PS/BN composites with a 30 wt% BN loading, the latter of which exhibited similar thermal conductivities to the molded composites. For PS/BN composites, BN fillers are difficult to disperse uniformly in the PS matrix due to poor compatibility. Therefore, large agglomerates of BN fillers inevitably existed in composites and resulted in a reduction in tensile strength. The higher concentration of BN fillers caused more agglomerates to form in the composites, resulting in a lower tensile strength. As a result, the control samples, characterized by uneven filler distribution, exhibited reduced tensile strengths compared to the molded composites. Similarly, the PS/BN composite with a 30 wt% BN loading showed a decreased tensile strength relative to the molded composites, despite their comparable thermal conductivities. Conversely, the molded PS/BN composites demonstrated superior tensile strength, attributable to their low filler content and homogeneous filler dispersion arising from the molding process. This evidence strongly suggests that compression molding represents a more advantageous strategy for achieving polymer composites with enhanced thermal conductivity, as opposed to merely increasing filler content.

The cross-sectional morphology of the molded PS/BN composites prepared herein was meticulously examined to assess the distribution and orientation of BN within the matrix subsequent to compression molding. Figure 3 reveals that all molded PS/BN composites exhibit enhanced in-plane orientation of BN or display in-plane cavities indicative of BN extraction. The presence of N elements in EDX images demonstrated the distribution of BN marked in SEM images. When subjected to consistent molding temperature and pressure, BN particles were observed to progressively adopt a “flattened” orientation from various directions as the number of molding cycles increased. This phenomenon suggests that the iterative process of folding and molding above the glass transition temperature of PS allows BN particles to migrate within the PS matrix, ultimately achieving a flat-sheet orientation under the applied molding pressure. However, the orientation of BN within the PS matrix showed no significant improvement with increased molding temperatures, provided the molding pressure and number of cycles remained constant. This could be attributed to elevated temperatures enhancing the mobility of the PS matrix, which, in turn, disrupts the orientation process of BN particles. Compared to the control sample, all molded samples demonstrated a pronounced aligned orientation of BN, in stark contrast to the substantial aggregation observed in the control. The molding process notably enhanced the dispersion of BN within the PS matrix, with no evident agglomeration detected in any of the molded samples. 

To quantitatively evaluate the degree of orientation of BN within the PS matrix, the XRD patterns of the PS/BN composites were analyzed. This analysis is predicated on the observation that different diffraction patterns emerge when X-rays interact with BN arranged in various orientations [9,35,46,47]. As depicted in Figure 4, the peak located at approximately 2θ = 27° corresponds to the (002) plane of BN, primarily aligned parallel to the composite surface (in-plane direction). Conversely, the peak around 2θ = 41.5° is attributed to the (100) plane of BN, which is oriented perpendicular to the surface of the sample (cross-plane direction). 

The orientation degree of BN within the PS matrix can be quantitatively assessed by analyzing the intensity ratio of I_(002)_/I_(100)_. All PS/BN composites, including the control samples that were not subjected to the molding process, displayed a pronounced peak for the (002) plane and a less intense peak for the (100) plane. This pattern suggests a natural propensity for the sheet-like BN fillers to orient within the polymer matrix, likely influenced by gravitational forces. The introduction of compression molding further heightened the intensity of the (002) plane, signifying an enhanced alignment of BN fillers along the in-plane direction. Notably, the value of I_(002)_/I_(100)_ directly correlated with the duration of molding, indicating a progressive increase in filler orientation. For instance, in Figure 4b, the I_(002)_/I_(100)_ ratio escalated to 1390 for PS/BN-C_5_ and 1340 for PS/BN-D_5_, showcasing a significant orientation improvement. The influence of molding temperature on filler orientation was also evident, with an increase in I_(002)_/I_(100)_ observed alongside rising molding temperatures given constant molding cycles and pressures. This trend can be attributed to the decreased viscosity of the PS matrix at higher temperatures, facilitating the orientation of BN fillers under applied molding pressure. However, the effect of temperature on orientation became negligible in PS/BN composites subjected to five or more compression molding cycles, as evidenced by the similar I_(002)_/I_(100)_ values observed in PS/BN-C_5_ and PS/BN-D_5_. Beyond a certain number of molding cycles, molding pressure exerted minimal influence on the orientation of BN within the matrix. However, an elevated molding pressure during five cycles of compression molding was associated with a higher I_(002)_/I_(100)_ ratio, indicative of a more pronounced filler orientation. The orientation of BN fillers plays a pivotal role in establishing effective thermal pathways within the composite, significantly mitigating phonon scattering and reducing the interfacial thermal resistance between PS and BN. Consequently, a well-oriented BN filler distribution contributes to an appreciable increase in thermal conductivity along the in-plane direction, underscoring the efficacy of compression molding in enhancing the thermal-management capabilities of PS/BN composites.

The thermal conductivities of the PS/BN composites, as presented in Figure 5, underscore the significant impact of BN incorporation and processing techniques on enhancing thermal transport within polystyrene matrices. The baseline thermal conductivity of the pure PS matrix was measured at a modest 0.033 W/m·K. The addition of BN to create control PS/BN samples resulted in an increased thermal conductivity of 0.22 W/m·K. This enhancement, though substantial, was limited by challenges in achieving uniform dispersion of BN within the PS matrix, which hindered the formation of an effective thermal pathway, even with a 10 wt% BN loading. The application of compression molding to the PS/BN composites significantly improved BN dispersion and orientation, facilitating the creation of thermal pathways. Consequently, the thermal conductivities of the molded PS/BN composites surpassed those of the control PS/BN samples, particularly in terms of in-plane thermal conductivity (Figure 5(a_1_)–(a_4_)). For instance, after a single compression molding cycle, the thermal conductivities of PS/BN-A_1_, PS/BN-B_1_, PS/BN-C_1_, and PS/BN-D_1_ were measured at 0.7, 1.0, 1.2, and 1.1 W/m·K, respectively, representing at least a threefold increase over the control samples. It is noteworthy that the enhancement of the in-plane thermal conductivity did not extend to the out-of-plane thermal conductivity, with all molded PS/BN composites exhibiting similar out-of-plane thermal conductivities to the control samples. Further analysis revealed that an increased number of molding cycles positively influenced the thermal conductivity of the molded PS/BN composites, with the highest thermal conductivity observed in samples subjected to five compression molding cycles, independent of the molding temperature and pressure used (Figure 5b). Among these, the PS/BN-C_5_ samples achieved an in-plane thermal conductivity of 1.6 W/m·K. Moreover, a higher molding pressure, particularly at 140 °C, was associated with increased thermal conductivity, indicating that optimal thermal conductivity was achieved for PS/BN composites molded at 140 °C compared to those processed at alternative temperatures (Figure 5c). This correlation between enhanced in-plane thermal conductivity and the degree of BN orientation within the PS matrix highlights the effectiveness of the molding process in achieving high thermal conductivity with only 10 wt% BN loading. These findings suggest that compression molding offers a promising approach for the scalable production of anisotropic thermally conductive polymer composites, optimizing thermal performance while maintaining minimal filler content.

Figure 6 presents thermal imaging results to elucidate the heat dissipation capabilities of the PS/BN composites, offering practical insights into the thermal-management performance of these materials. The experiment involved heating the samples to 60 °C and subsequently monitoring their cooling process upon removal from the heat source. The cooling rate of the PS/BN composites was directly relevant to the orientation of BN within the PS matrix. Specifically, the PS/BN-C_5_ composite, which exhibited the highest degree of BN orientation, also displayed the fastest cooling rate among the samples tested, in addition to the highest thermal conductivity. Further analysis linked to the thermal imaging, as demonstrated in Figure 6b, revealed that the surface temperature of the PS/BN-C_5_ sample decreased rapidly, reaching a significantly lower temperature within approximately 30 s. In contrast, the pure PS matrix took over 60 s to cool to the same temperature, with the control PS/BN sample and the PS/BN-B_1_ sample showing intermediate cooling times. This indicates the superior thermal dissipation performance of PS/BN-C_5_ compared to the other samples evaluated. Additionally, when the samples were placed on a bench heated to 60 °C, they underwent a heating process from room temperature to 60 °C. Remarkably, the PS/BN-C_5_ composite showed the quickest temperature increase rate, with a nearly 20 °C rise within 120 s, signifying its exceptional in-plane thermal conductivity. Conversely, the central temperature of the pure PS matrix experienced only a 7 °C increase after being placed on the 60 °C hot bench. These findings clearly demonstrate that the PS/BN composites, particularly those fabricated through five cycles of compression molding at 140 °C under a pressure of 10 MPa, achieved the highest in-plane thermal conductivity and exhibited optimal thermal management performance. This underscores the significant role of BN orientation in enhancing the thermal response of polymer composites, highlighting the effectiveness of compression molding in improving the thermal-management capabilities of PS/BN composites.

## 4. Conclusions

In conclusion, this study successfully developed PS/BN composites exhibiting enhanced thermal conductivities through a straightforward compression molding technique. The implementation of this molding process was instrumental in aligning BN fillers within the in-plane direction of the PS matrix, effectively establishing a thermally conductive network with a modest BN loading of 10 wt%. Notably, the molded PS/BN composites, even after a single molding cycle, demonstrated a threefold enhancement in thermal conductivity compared to the control samples. Furthermore, incremental increases in molding duration significantly improved the orientation of BN fillers in the in-plane direction, culminating in a marked elevation of in-plane thermal conductivity. By modulating the molding temperature and pressure, it was possible to fine-tune both the alignment of BN within the composite and its thermal conductivity. Specifically, PS/BN composites subjected to five molding cycles at 140 °C under a pressure of 10 MPa achieved an in-plane thermal conductivity of 1.6 W/m·K, without compromising mechanical integrity. The methodologies outlined in this research underscore the feasibility of using compression molding to fabricate anisotropic thermal management materials on a large scale, offering substantial promise for the advancement of thermal conductivity in polymer-based composites.

## Figures and Tables

**Figure 1 polymers-16-01169-f001:**
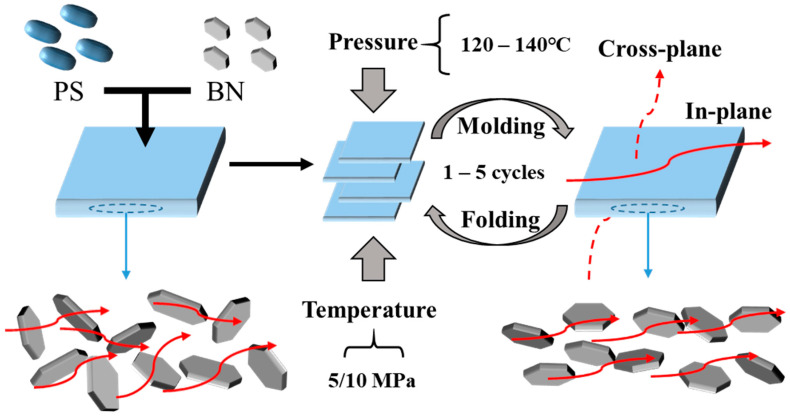
Schematic illustration of molded PS/BN composites with their thermal conductivity enhanced through a molding technique(Red arrows indicate the direction of heat flow in the schematic diagram).

**Figure 2 polymers-16-01169-f002:**
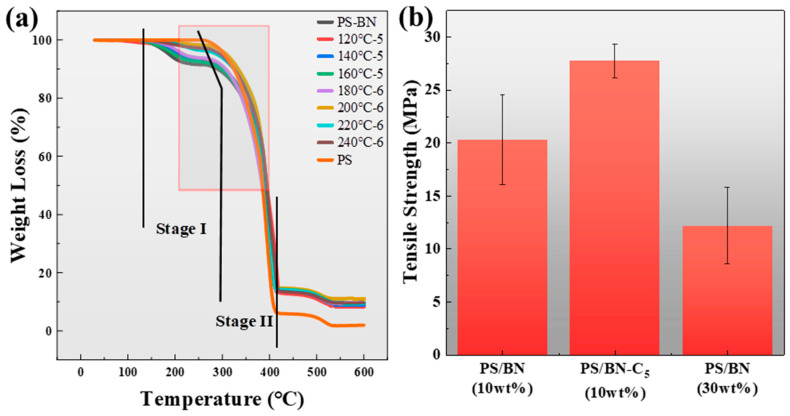
TGA curves (**a**) and tensile strength (**b**) of PS/BN composites.

**Figure 3 polymers-16-01169-f003:**
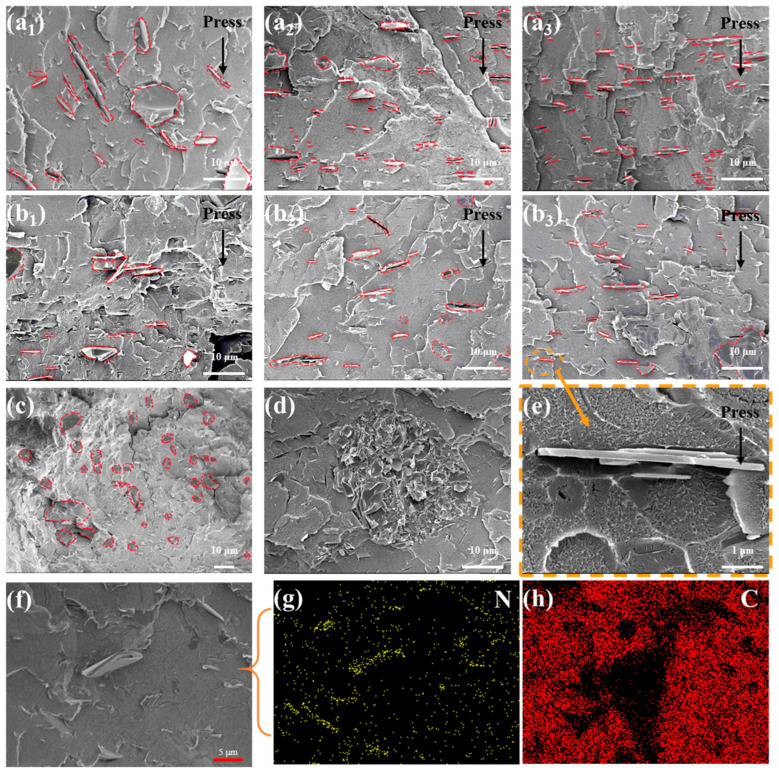
Cross-sectional SEM images of PS/BN composites: (**a_1_**–**a_3_**) PS/BN-A_1,3,5_; (**b_1_**–**b_3_**) PS/BN-B_1,3,5_ (BN is marked with red dashed lines); (**c**,**d**) control PS/BN sample; (**e**) Zoomed-in image of PS/BN-B_3_; (**f**) PS/BN-C_1_; (**g**,**h**) EDX images of (**f**).

**Figure 4 polymers-16-01169-f004:**
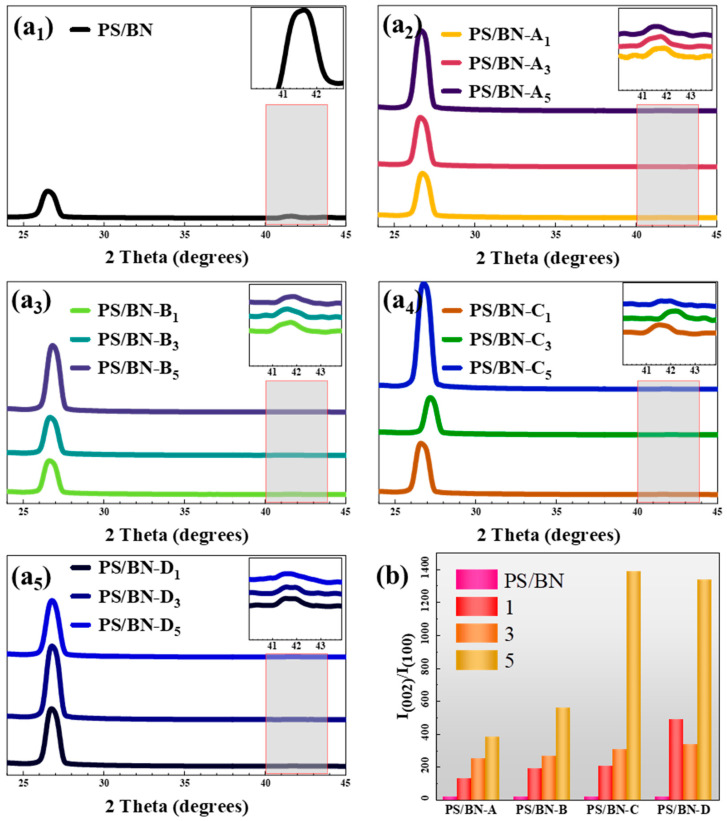
(**a**) The XRD patterns (Samples with different molding parameters are represented as (**a_1_**–**a_5_**)) and (**b**) the values of I_(002)_/I_(100)_ of the PS/BN composites.

**Figure 5 polymers-16-01169-f005:**
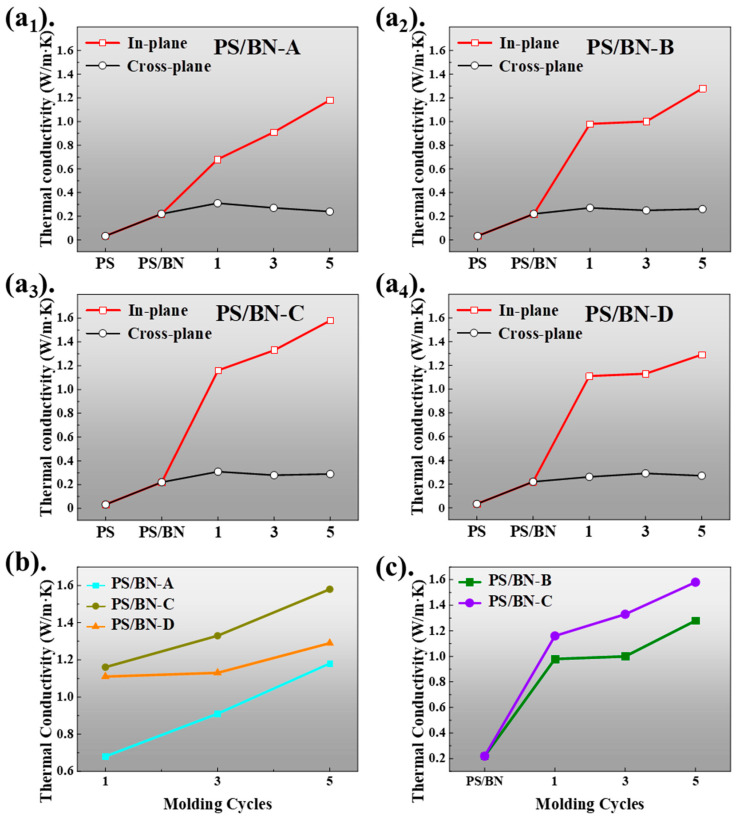
The thermal conductivities of PS/BN composites. (**a_1_**–**a_4_**) The in-plane/cross-plane thermal conductivity of samples with different molding parameters; (**b**) The in-plane thermal conductivity of samples at different molding temperature; (**c**) The in-plane thermal conductivity of samples under different molding pressures.

**Figure 6 polymers-16-01169-f006:**
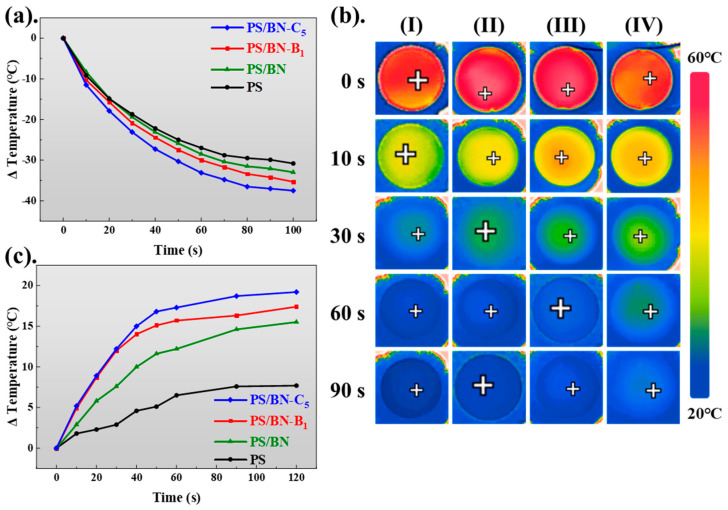
(**a**–**c**) The cooling rates, infrared images, and heating rates of PS/BN composites and PS matrix. Sample I refers to PS/BN-C_5_ samples, while samples II–IV correspond to PS/BN-B_1_, control PS/BN samples, and pure PS, respectively.

**Table 1 polymers-16-01169-t001:** Molding parameters of PS/BN composites.

Sample	Molding Parameters
PS/BN-A_1,3,5_	120 °C/10 MPa
PS/BN-B_1,3,5_	140 °C/5 MPa
PS/BN-C_1,3,5_	140 °C/10 MPa
PS/BN-D_1,3,5_	160 °C/10 MPa
PS/BN	Solution casting

Note: 1, 3, and 5 refer to molding cycles with similar molding parameters.

## Data Availability

The data presented in this study are available upon request from the corresponding author (due to privacy).

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
