# Peer review of "Enhancing Thermal Conductivity in Polymer Composites through Molding-Assisted Orientation of Boron Nitride"

_polymers, 2024, doi:10.3390/polym16081169_

Round 1
Reviewer 1 Report
Comments and Suggestions for Authors
The manuscript describes the mechanical properties of composite materials made from boron nitride and polystyrene. The manuscript can be accepted for publication after minor revisions, as under:
1. Please improve the language of the manuscript.
2. The authors have used old references Please update the literature with respect to the recent references
3. The units of different physical quantities should be corrected, as the last number must be superscript.
4. Why does the authors optimise the concentration of BN as it may provide in depth insights related to the heat dissipation capacity (Fig. 6)
Comments on the Quality of English LanguageThe manuscript is well-written with good quality English. Just need slight polishing
Author Response
Dear Editor and Reviewer
Thank you for your letter and for the reviewers’ comments concerning our manuscript entitled “Enhancing Thermal Conductivity in Polymer Composites through Molding-Assisted Orientation of Boron Nitride” (No. polymers-2956409). Those comments are all valuable and very helpful for revising and improving our paper. We have studied comments carefully and have made corrections which we hope meet with approval. The revised portion is marked in red in the revised paper.
Detailed responses to the reviewers’ comments are as follows.
The manuscript describes the mechanical properties of composite materials made from boron nitride and polystyrene. The manuscript can be accepted for publication after minor revisions, as under:
- Please improve the language of the manuscript.
Answer: As suggested, the language has been improved in the revised manuscript.
- The authors have used old references Please update the literature with respect to the recent references
Answer: As suggested, the recent references have been supplemented.
- The units of different physical quantities should be corrected, as the last number must be superscript.
Answer: As suggested, the physical quantities have been corrected in the revised manuscript.
- Why does the authors optimise the concentration of BN as it may provide in depth insights related to the heat dissipation capacity (Fig. 6)
Answer: We are sorry for the confusion caused by the inadequate expression of samples. In the revised manuscript, we have modified these confusing samples in Figure 6 caption. The current research aims to enhance the orientation of BN in the PS matrix by compression molding to improve the thermal conductivity of polymer composites with a low filler loading. Therefore, all the molded samples have a BN loading of 10 wt%. We did not investigate the effect of BN concentration on the thermal conductivity of PS/BN composites. We deeply appreciate the reviewer's comments and would very much like to consider this direction of research in our future work.
We tried our best to improve the manuscript and made some changes in the manuscript. These changes will not influence the content and framework of the paper. We hope that the correction will meet with approval. If further actions are required, please let us know.
Again, thank you very much for your comments and suggestions.
Best wishes,
Huarong Nie
Key Laboratory of Rubber-Plastics, Ministry of Education/Shandong Provincial Key Laboratory of Rubber-Plastics, School of Polymer Science and Engineering, Qingdao University of Science and Technology, Qingdao 266042, China
Tel: +86-0532-84022951, Fax: +86-0532-84022951
E-mail: niehr@iccas.ac.cn

Reviewer 2 Report
Comments and Suggestions for Authors
The authors demonstrated the development of PS/BN composites exhibiting thermal conductivities improved through a compression molding technique. They tested molding temperatures, pression and cycles to fully understand the effects on the thermal conductivity of the composites. Some comments can be found below:
The authors explain that polymers are intrinsically insulators and would need the increment of a thermally conductive material to became thermally conductive, able to be used in electronics. But, in page 2, authors said that the “…inclusion of thermally conductive insulating fillers preserves the outstanding insulating properties of polymers…’”. It is confusing.
What kind of “intricate chemical reactions” are needed to modify the surface of the fillers?
Did the authors measured the thickness of the samples, once it is important to maintain it uniform across the samples?
In figure 3, for molding temperatures below 180 °C, there is a weight loss of the samples significantly different from the ones molded at temperatures above 180 °C at 300 °C.
The authors claimed that “…the initial decomposition temperature of the PS/BN control sample was observed to be marginally higher than that of pure PS”. Where is the data? How did the authors come to this conclusion?
Figure 2 (a) Y-axis should read “weight loss”.
Do the authors have a hypothesis on why the out-of-plane thermal conductivities are similar to the one of the control samples?
Author Response
Dear Editor and Reviewer
Thank you for your letter and for the reviewers’ comments concerning our manuscript entitled “Enhancing Thermal Conductivity in Polymer Composites through Molding-Assisted Orientation of Boron Nitride” (No. polymers-2956409). Those comments are all valuable and very helpful for revising and improving our paper. We have studied comments carefully and have made corrections which we hope meet with approval. The revised portion is marked in red in the revised paper.
Detailed responses to the reviewers’ comments are as follows.
The authors demonstrated the development of PS/BN composites exhibiting thermal conductivities improved through a compression molding technique. They tested molding temperatures, pression and cycles to fully understand the effects on the thermal conductivity of the composites. Some comments can be found below:
- The authors explain that polymers are intrinsically insulators and would need the increment of a thermally conductive material to became thermally conductive, able to be used in electronics. But, in page 2, authors said that the “…inclusion of thermally conductive insulating fillers preserves the outstanding insulating properties of polymers…’”. It is confusing.
Answer: In the revised manuscript, we have modified these confusing sentences.
- What kind of “intricate chemical reactions” are needed to modify the surface of the fillers?
Answer: Various surface modifications of fillers have been performed to improve the compatibility between the fillers and the polymer matrix. The reported methods include acid treatment, basic treatment, grafting polymer brushes, and so on. The relevant references have been cited in the revised manuscript.
- Did the authors measured the thickness of the samples, once it is important to maintain it uniform across the samples?
Answer: The mold thickness for sample preparation was 0.5 mm. Under the molding pressure, all the samples have a uniform thickness. In addition, we measured the sample thickness at three randomly selected locations with a gauge to achieve the average value before testing the thermal diffusion coefficient.
- In figure 3, for molding temperatures below 180 °C, there is a weight loss of the samples significantly different from the ones molded at temperatures above 180 °C at 300 °C.
Answer: Indeed, one stage of weight loss was observed for PS sample, which is directly relevant to the decomposition of PS. However, there are two stages of weight loss for PS/BN composites. The first stage involved the removal of impurities and solvent residues, and the second stage started from 270 ℃, corresponding to the decomposition of PS. Compared to those samples molded at a temperature below 180 ℃, the samples that were molded at a higher temperature and experienced more molding cycles exhibited less weight loss, which can be attributed to the removal of more impurities or solvent residues during the molding process. At 300 ℃, all samples exhibited the degradation of the PS matrix. However, the molded samples at a temperature above 180 ℃ exhibited less weight loss because of the improved thermal stability with the improved orientation of BN.
- The authors claimed that “…the initial decomposition temperature of the PS/BN control sample was observed to be marginally higher than that of pure PS”. Where is the data? How did the authors come to this conclusion?
Answer: Thank the reviewer for the kind reminder. In the revised manuscript, we supplement the TGA curve of the pure PS sample. A distinct weight loss started from 250 ℃ for the decomposition of PS while a significant weight loss started from 270 ℃ for the decomposition of PS in PS/BN composites EDX
- Figure 2 (a) Y-axis should read “weight loss”.
Answer: As suggested, the Y-axis of Figure 2a has been corrected.
- Do the authors have a hypothesis on why the out-of-plane thermal conductivities are similar to the one of the control samples?
Answer: For PS/BN composites, all exhibited a relatively low out-of-plane thermal conductivity irrespective of the molding parameters. The reason can be attributed to that the molding process allows the improved orientation of BN filler in the in-plane direction. The distribution of BN in the out-of-plane direction is as random as that of the PS/BN control samples. Compared to PS/BN control samples, the molded samples had the repeated folding process during molding cycles, which enabled the arrangement of BN in a more orderly manner along the out-of-plane direction, resulting in relatively high out-of-thermal conductivity.
We tried our best to improve the manuscript and made some changes in the manuscript. These changes will not influence the content and framework of the paper. We hope that the correction will meet with approval. If further actions are required, please let us know.
Again, thank you very much for your comments and suggestions.
Best wishes,
Huarong Nie
Key Laboratory of Rubber-Plastics, Ministry of Education/Shandong Provincial Key Laboratory of Rubber-Plastics, School of Polymer Science and Engineering, Qingdao University of Science and Technology, Qingdao 266042, China
Tel: +86-0532-84022951, Fax: +86-0532-84022951
E-mail: niehr@iccas.ac.cn

Reviewer 3 Report
Comments and Suggestions for Authors
This study successfully fabricated a polystyrene (PS)/boron nitride (BN) composite with elevated thermal conductivity using 10 wt% BN content through compression molding. The optimized molding process resulted in in-plane alignment of BN within the PS matrix, leading to a threefold enhancement in thermal conductivity. The highest thermal conductivity achieved was 1.6 W/m⸱K in PS/BN composites subjected to five molding cycles at 140 ℃ and 10 MPa, without compromising mechanical properties. These findings highlight the efficacy of compression molding for producing polymer composites with improved thermal conductivity. Indeed, it is interesting study and I would like to recommend it for publication after the careful revisions as suggested below.
(1). The novelty highlight of the work is somewhat missing in the current version. It is suggested to enhance the introduction section by citing the latest relevant literature reports. The following articles discussing about properties of polymeric nanocomposites should be read and cited in the revised manuscript. https://doi.org/10.1007/s00170-017-0764-5
At the end of the introduction section, a few lines of the achieved thermal properties of polymer-BN composites in the current study should be mentioned for easy reading by readers.
(2). In the experimental section, the purity of the chemicals and materials used (e.g., hexagonal boron nitride (h-BN)) should be mentioned. Also, the detailed experimental procedure should be mentioned to reproduce these results. This is a very important aspect of research for other researchers/industries working in a similar direction.
(3). In the schematic fabrication (Figure. 1), please quantify the temperature (C), pressure (Pa), time, etc. for a better understanding of composite preparation for the readers.
(4). The stages of weight loss (%) in the TGA cureves should be marked in figure 2a. Also, please mention the wt.% of PS/BN-C5 in the figure 2b. Could you please discuss the reason for the highest value of tensile strength (MPA) of the PS/BN-C5 composite in the revised manuscript.
(5). Could you please explain why the increase of BN from 10 wt% to 30 wt% has negative impact on the tensile strength. Also, what should be the optimum value of BN in the composite material to obtain the highest UTS ?
(6). Please provide the EDX elemental mapping along with the SEM results for better understanding.
(7). The TEM, STEM and SAED images should be provided (if possible) in the revised manuscript to clearly confirm the morphology of h-BN after being incorporated into composite material.
(8). The inset of Figure 4a (XRD pattern) is not visible; could you please provide a better quality image? These XRD patterns were plotted after normalization or without normalization. It is very important to mention for better comparison of peak intensities.
(9). Please discuss the cooling rate effect on the morphological evolution of h-BN (if any). Also, the procedure and experimental conditions of infrared imaging should be discussed in the experimental section.
(10). All the references should be according to journal format.
Comments on the Quality of English Language
There are several grammar mistakes. Thus, overall, the English language should be improved.
Author Response
Dear Editor and Reviewer
Thank you for your letter and for the reviewers’ comments concerning our manuscript entitled “Enhancing Thermal Conductivity in Polymer Composites through Molding-Assisted Orientation of Boron Nitride” (No. polymers-2956409). Those comments are all valuable and very helpful for revising and improving our paper. We have studied comments carefully and have made corrections which we hope meet with approval. The revised portion is marked in red in the revised paper.
Detailed responses to the reviewers’ comments are as follows.
This study successfully fabricated a polystyrene (PS)/boron nitride (BN) composite with elevated thermal conductivity using 10 wt% BN content through compression molding. The optimized molding process resulted in in-plane alignment of BN within the PS matrix, leading to a threefold enhancement in thermal conductivity. The highest thermal conductivity achieved was 1.6 W/m⸱K in PS/BN composites subjected to five molding cycles at 140 ℃ and 10 MPa, without compromising mechanical properties. These findings highlight the efficacy of compression molding for producing polymer composites with improved thermal conductivity. Indeed, it is interesting study and I would like to recommend it for publication after the careful revisions as suggested below.
(1) The novelty highlight of the work is somewhat missing in the current version. It is suggested to enhance the introduction section by citing the latest relevant literature reports. The following articles discussing about properties of polymeric nanocomposites should be read and cited in the revised manuscript. https://doi.org/10.1007/s00170-017-0764-5
At the end of the introduction section, a few lines of the achieved thermal properties of polymer-BN composites in the current study should be mentioned for easy reading by readers.
Answer: As suggested, the recent references, including the recommended reference, have been supplemented in the revised manuscript. Moreover, the thermal properties of the as-prepared composites have been provided at the end of the introduction section.
(2) In the experimental section, the purity of the chemicals and materials used (e.g., hexagonal boron nitride (h-BN)) should be mentioned. Also, the detailed experimental procedure should be mentioned to reproduce these results. This is a very important aspect of research for other researchers/industries working in a similar direction.
Answer: As suggested, detailed information on BN has been provided. Moreover, the detailed experimental procedure has been mentioned in the revised manuscript.
(3) In the schematic fabrication (Figure. 1), please quantify the temperature (C), pressure (Pa), time, etc. for a better understanding of composite preparation for the readers.
Answer: As suggested, the molding parameters have been provided in Figure 1.
(4) The stages of weight loss (%) in the TGA curves should be marked in figure 2a. Also, please mention the wt.% of PS/BN-C5 in the figure 2b. Could you please discuss the reason for the highest value of tensile strength (MPA) of the PS/BN-C5 composite in the revised manuscript.
Answer: Thank the reviewer for the valuable comments. The corresponding markers have been provided in the revised manuscript. In Figure 2a, a distinct weight loss occurred at 250 ℃ for the decomposition of PS. However, there are two stages of weight loss for PS/BN composites. The first stage of weight loss referred to the removal of more impurities or solvent residues in PS/BN composites, and the second stage of weight loss strated from 270 ℃ corresponds to the decomposition of PS in PS/BN composites.
It is difficult for BN fillers to disperse uniformly in the PS matrix due to poor compatibility. Therefore, the large agglomerates of BN fillers inevitably existed in PS/BN composites and resulted in the reduction of tensile strength. The higher concentration of BN fillers caused more agglomerates to form in composites, resulting in a lower tensile strength. Correspondingly, the PS/BN control sample with a 10 wt% BN loading exhibited higher tensile strength than PS/BN composites with a 30 wt % BN loading. For the PS/BN molded samples having a 10 wt% BN loading, the molding process allows an improved dispersion of BN in composites, thus a higher tensile strength is observed. As a result, PS/BN-C5 exhibited the highest value of tensile strength (MPA) in this case. We have described these in the revised manuscript.
(5). Could you please explain why the increase of BN from 10 wt% to 30 wt% has negative impact on the tensile strength. Also, what should be the optimum value of BN in the composite material to obtain the highest UTS ?
Answer: It is difficult for BN fillers to disperse uniformly in the PS matrix due to poor compatibility. Therefore, the large agglomerates of BN fillers inevitably existed in PS/BN composites and resulted in the reduction of tensile strength. The higher concentration of BN fillers caused more agglomerates to form in composites, resulting in a lower tensile strength. The current research aims to enhance the orientation of BN in PS by compression molding to improve the thermal conductivity of polymer materials at a low filler loading. Therefore, all the molded samples have a BN loading of 10 wt%. We did not investigate the effect of BN concentration on the tensile strength and thermal conductivity of PS/BN composites. Therefore, it is difficult for us to tell the optimum value of BN in the composite material to obtain the highest UTS. We deeply appreciate the reviewer's comment and would very much like to consider this direction of research in our future work.
(6). Please provide the EDX elemental mapping along with the SEM results for better understanding.
Answer: Thank the reviewer for the kind suggestion. The EDX images have been provided in the revised manuscript. As shown in the following figures, the EDX images were recorded to distinguish the location and distribution of BN fillers in the PS matrix. The presence of N elements in EDX images reveals the distribution of BN marked in SEM images.
(7). The TEM, STEM and SAED images should be provided (if possible) in the revised manuscript to clearly confirm the morphology of h-BN after being incorporated into composite material.
Answer: This is a good suggestion. Since the SEM images displayed the distribution and orientation of BN in the PS matrix, other characterizations were not performed on the PS/BN composites in the manuscript. Additionally, it is difficult for us to get STEM, SAED, and even TEM images due to limitations in the experimental conditions.
(8). The inset of Figure 4a (XRD pattern) is not visible; could you please provide a better quality image? These XRD patterns were plotted after normalization or without normalization. It is very important to mention for better comparison of peak intensities.
Answer: As suggested, a more clear XRD pattern has been provided in Figure 4. All the XRD patterns were plotted after normalization to better compare the peak intensities.
(9). Please discuss the cooling rate effect on the morphological evolution of h-BN (if any). Also, the procedure and experimental conditions of infrared imaging should be discussed in the experimental section.
Answer: As suggested, the procedure and experimental conditions of infrared imaging have been supplemented in the experimental section. In this study, the investigated molding parameters include the molding temperature, cycle numbers, and pressure. All molded samples were cooled rapidly to maintain the distribution of BN after the PS/BN composites underwent compression molding with the designed parameters. We deeply appreciate the reviewer's comment and would very much like to consider this direction of research in our future work. The cooling rates in Figure 6 were recorded to compare the heat dissipation capabilities of PS/BN composites after they were removed from a bench heated up to 60 ℃. The sample with a higher thermal conductivity displayed a higher cooling rate.
(10). All the references should be according to journal format.
Answer: All of the references have been revised according to the journal format.
We tried our best to improve the manuscript and made some changes in the manuscript. These changes will not influence the content and framework of the paper. We hope that the correction will meet with approval. If further actions are required, please let us know.
Again, thank you very much for your comments and suggestions.
Best wishes,
Huarong Nie
Key Laboratory of Rubber-Plastics, Ministry of Education/Shandong Provincial Key Laboratory of Rubber-Plastics, School of Polymer Science and Engineering, Qingdao University of Science and Technology, Qingdao 266042, China
Tel: +86-0532-84022951, Fax: +86-0532-84022951
E-mail: niehr@iccas.ac.cn

Round 2
Reviewer 1 Report
Comments and Suggestions for Authors
Accept
Reviewer 2 Report
Comments and Suggestions for Authors
The manuscript can be published in the present form.
Author Response
We greatly appreciate the time the reviewer spent in reviewing our manuscript. We feel great thanks for your accepting our manuscript.
Reviewer 3 Report
Comments and Suggestions for Authors
Thank you for your time and efforts
Author Response

(The authors gave the same response as above.)
